# Multiple-Beam Steering Using Graphene-Based Coding Metasurfaces

**DOI:** 10.3390/mi14051018

**Published:** 2023-05-09

**Authors:** Bin Zheng, Xin Rao, Yuyu Shan, Chuandong Yu, Jingke Zhang, Na Li

**Affiliations:** The Key Laboratory of Electronic Equipment Structure Design, Ministry of Education, Xidian University, Xi’an 710071, China; 20041110182@stu.xidian.edu.cn (B.Z.); 21041211877@stu.xidian.edu.cn (X.R.); 21041212129@stu.xidian.edu.cn (Y.S.); 22041212841@stu.xidian.edu.cn (C.Y.); 21041212114@stu.xidian.edu.cn (J.Z.)

**Keywords:** antenna, beam manipulation, coding metasurfaces, graphene

## Abstract

Recently, the coding metasurface has gained significant attention due to its exceptional potential in controlling electromagnetic (EM) waves with the rapid development of wireless communication systems. Meanwhile, graphene shows tremendous promise for the implementation of reconfigurable antennas due to its high tunable conductivity and its unique property that makes it a very suitable material for realizing steerable coded states. In this paper, we first propose a simple structured beam reconfigurable millimeter wave (MMW) antenna using a novel graphene-based coding metasurface (GBCM). Different from the previous method, its coding state can be manipulated by altering the sheet impedance of graphene instead of bias voltage. Then, we design and simulate several most popular coding sequences, including dual-, quad-and single-beam-generated implement, 30° beam deflection, as well as a random coding sequence for radar cross-section (RCS) reduction. The theoretical and simulation results show that graphene has great potential for MMW manipulation applications, which lay a foundation for the subsequent development and fabrication of GBCM.

## 1. Introduction

The capability of manipulating electromagnetic (EM) waves, including the frequency [1], phase [2], magnitude [3], polarization [4], direction of propagation [5], and wave patterns [6], is important in wireless communication technologies. In recent years, metasurface (MS), due to the characteristic of the phase discontinuities across its subwavelength metallic surface, has the ability to control the reflected or transmitted beams and attracted the wide attention of researchers [7]. Since Cui et al. [8] proposed the concept of coding metamaterials in 2014, the MS has ushered in the golden age of theoretical and practical research, which offers a new strategy to control the EM wave in an arbitrary direction by designing the coding phase distribution sequences. In particular, by encoding different MS sequences, a variety of EM wave functions can be realized, such as multi-beam forming [9], beam deflection [10], orbital angular momentum (OAM) generation [11], radar cross-section (RCS) reduction [12] and other applications.

The methodology of the coding beam reconfiguration needs a stable amplitude response and periodic gradient phase shifts in the range of 0–2π [13,14,15,16]. Conventional metallic MS form phase differences by changing the sizes [13], orientations [14,15,16], or shapes [5,10] of the unit cells. Particularly, encoding two types of unit cells with 0 and π phase responses as “0” and “1” elements can realize a 1-bit coding MS [8,9]. The 2-bit coding states were realized by four types of unit cells which have phase responses of 0, π/2, π, and 3π/2 to mimic “00”, “01”, “10”, and “11” elements, respectively. Higher-bit coding elements can realize more precise beam control. However, with the rapid development of millimeter wave (MMW) and terahertz wave (THz) bands in recent years, the traditional metallic MS is no longer suitable due to a decrease in conductivity and skin depth [17]. On the other hand, to realize the dynamic encoding of the state of the coding MS, diodes [18] were used to switch unit cells. However, these kinds of lumped components have complex structures, long response times, and are difficult to manufacture at high frequencies.

In recent years, graphene has provided more abundant manipulation means for the regulation of EM waves by taking advantage of its tunable conductivity [19,20]. Meanwhile, numerous functions of graphene manipulation of EM waves have been exploited and reported, such as THz absorber [21], polarization controller [22], adjustable graphene absorber [23], sensor [24], etc. In particular, its unique properties in electronic and optical properties enable graphene as a very suitable material for achieving manipulation of the coding state. According to the Kubo formula [20,25,26,27,28,29,30,31,32,33,34], we can notice that the conductivity of graphene can be dynamically controlled by altering frequency, chemical potential, relaxation time, and temperature. Generally speaking, in simulation experiments, the dynamic regulation of graphene is achieved by setting different chemical potentials [20,25,26,27], Fermi levels [28,29,30,31], or bias voltages [32,33]. However, all three methods essentially alter graphene’s EM properties by adding bias circuits, which can lead to complex structures that are difficult to manufacture with existing technology. Therefore, graphene-based coding metasurface (GBCM) antennas are mostly in the simulation and numerical calculation stage. Fortunately, paper [34] fabricated and measured beam reconfiguration graphene antenna in microwaves by taking advantage of the tunable sheet impedance of graphene. However, it has a single function and does not incorporate coding techniques to achieve more precise beam control.

In this paper, for the first time, we proposed a new approach for the EM wave beam regulation with complex functions by combining the tunable sheet impedance of graphene with the coding MS, as is shown in Figure 1. To the best of the authors’ knowledge, beam reconfiguration based on GBCM has not been studied yet at MMW. We are looking forward to introducing and generalizing the sheet impedance tuning method and employing it for designing GBCM to enrich the application of graphene antennae. In this work, five different coding sequences have been simulated, implementing dual-, quad- and single-beam generated beam deflection (30°), as well as a random coding sequence for radar cross-section (RCS) reduction. The results show that the beam steering can be realized by changing the graphene sheet resistance and combining it with the MS coding principle, which lays a foundation for the subsequent fabrication of graphene antennas in the field of MMW wave application.

## 2. Materials and Methods

The design of any MS starts with its basic unit cells. For beam manipulation, a unit cell needs to be able to simultaneously satisfy the ability to control the phase response and generate high reflection amplitude in a wide band [6,20]. Therefore, we first need to clarify the EM properties of graphene. Then, the coding graphene unit cells were designed based on the regulation theory. Finally, several GBCM antennas are designed and simulated according to the phase gradient coding principle.

### 2.1. Electromagnetic Properties of Graphene

Nanomaterial graphene is a monolayer of carbon atoms arranged in a hexagonal lattice that can be modeled as an extremely thin conductive layer. The conductivity of graphene can be calculated using the Kubo formula [20,25,26,27,28,29,30,31,32], which can be approximated as:(1)$$\sigma (\omega ,\mu_{c},\Gamma ,T)=\frac{je^{2}\left(\omega -j2\Gamma \right)}{\pi \hbar^{2}}[\frac{1}{{\left(\omega -j2\Gamma \right)}^{2}}\int_{0}^{\infty }\varepsilon \left(\frac{\partial f_{d}\left(\varepsilon \right)}{\partial \varepsilon }-\frac{\partial f_{d}\left(-\varepsilon \right)}{\partial \varepsilon }\right)d\varepsilon -\int_{0}^{\infty }\frac{\partial f_{d}\left(-\varepsilon \right)-f_{d}\left(\varepsilon \right)}{{\left(\omega -j2\Gamma \right)}^{2}-4{\left(\varepsilon /\hbar \right)}^{2}}d\varepsilon ]$$
where $f_{d}\left(\varepsilon \right)$ is the Fermi Dirac distribution function:(2)$$f_{d}\left(\varepsilon \right)={{(e}^{\frac{{(\varepsilon -\mu }_{c})}{k_{B}T}}+1)}^{-1}$$

At the THz band, graphene-based devices are enabled solely by the intraband transition. The graphene conductivity can be approximately expressed as:(3)$$\sigma_{intra}(\omega ,\mu_{c},\Gamma ,T)=-j\frac{e^{2}K_{B}T}{\pi \hbar^{2}\left(\omega -j2\Gamma \right)}[\frac{\mu_{c}}{k_{B}T}+2\ln\left(e^{-\frac{\mu_{c}}{K_{B}T}}+1\right)]$$
where *j* is the imaginary unit, *e* = −1.6 × 10^−19^ C is the electron charge, ℏ = 1.05 × 10^−34^ Js is the reduced Plank’s constant, $k_{B}$ = 1.38 × 10^−23^ m^2^kg/s^2^K is Boltzmann’s constant, *T* = 300 *K* is the room temperature and τ is the relaxation time, Γ=1/2τ is the scattering rate, $\mu_{c}$ is the chemical potential. From the above equations, we can notice that the conductivity of the graphene can be dynamically controlled by altering frequency, chemical potential, relaxation time, and temperature. This property enables the reconfiguration ability in graphene-based devices.

At the microwave band, based on the Kubo formula in Equation (1), the resistance of graphene impedance remains almost constant, while the reactance term tends to 0 and could be neglected. Therefore, graphene can be approximately treated as a resistive sheet without dispersion [34] and modeled using ohmic sheet boundary in commercial software CST Microwave Studio. Meanwhile, many works devoted to the practical fabrication of graphene antennae have been reported recently [35,36]. In fabrication, the sheet resistance of single-layer graphene (SLG) can reach the range from 1000 to 1200 Ω/sq, and for multilayer graphene (MLG), the sheet resistance is in a range from 5 to 325 Ω/sq [34]. Due to the complex structure of the beam control method using bias voltage, this paper proposes a GBCMs design method using graphene with different sheet impedances for MMW beam steering.

### 2.2. Unit Cell Design

The unit cell of the proposed 1-bit graphene structure is illustrated in Figure 1, which is composed of four layers: graphene layer, polyvinyl chloride (PVC, *εr* = 3.5) substrate, Taconic TLY-5 substrate (*εr* = 2.2), and the copper layer as GND. Figure 2a shows the exact size of the graphene unit. The graphene patch consists of four small squares and is transferred onto a PVC layer with *h*1 = 0.075 mm height and placed on a Taconic TLY-5 substrate with *h*2 = 0.787 mm height. The spacing between the units should be a subwavelength to provide efficient scattering and prevent the occurrence of grating diffraction. However, it should not be too small; otherwise, the strong nearfield coupling between neighboring antennas would perturb the designed scattering amplitudes and phases [7]. In our work, we set the initial size of the metasurface unit p to 0.5 × λ_0_, where λ_0_ is the free space wavelength at 55 GHz. Then, it is optimized in the CST simulation software. The performance of the unit cell is calculated with the final optimized parameters as follows, *p* = 3.8 mm, *l*1 = 3.15 mm, *l*2 = 0.3 mm, *h*1 = 0.075 mm, and *h*2 = 0.787 mm. Graphene with a sheet resistance of Rs = 10 Ω/sq is coded as “0” and Rs = 1000 Ω/sq is coded as “1”. The proposed GBCM unit has been simulated in the CST Microwave studio [29].

Figure 2a shows the reflection amplitude of “0” and “1” elements with sheet resistance of 10 Ω/sq and 1000 Ω/sq. The reflection amplitudes of “0” and “1” elements are both 0.65 at the frequency from 40 GHz to 60 GHz. Figure 2b shows the reflection phase and phase difference of “0” and “1” elements with surface resistance of 10 Ω/sq and 1000 Ω/sq. Nearly the phase difference of 180° could be realized from 40 GHz to 60 GHz, which indicates the ability to produce large phase differences by changing the resistive component of graphene impedance. Based on this feature, different angles of reflected lobes could be achieved by properly arranging the graphene coding arrays.

### 2.3. Graphene-Based Coding MS Array Design

To illustrate that beam steering can be altered by varying the sheet resistance of graphene without changing the geometrical parameters of the structure, MSs, including *M × N* tunable units, are considered. In this case, we need to use the generalized reflection law to evaluate the response of the MS. The principle of abnormal reflection was first proposed by F. CAPASSO and his team at Harvard University in 2011 in the journal Science [7]. The forward direction of the EM beam can be controlled by introducing a linear propagation phase at the medium interface. The correspondence between phase gradient and angular deflection can be described by generalized Snell’s law:(4)$$n_{r}\sin\theta_{r}-n_{i}\sin\theta_{i}=\frac{\lambda_{0}}{2\pi }\frac{d\varphi }{dx}$$
where $\theta_{r}$ and $\theta_{i}$, respectively, represent the angle of reflection and incidence, $n_{i}$ represents the refractive index of the medium, dφ/dx represents the change of the phase of reflection per unit length, and *dx* is the length of the element. $\lambda_{0}$ represents the wavelength in vacuum corresponding to the EM wave at the operating frequency. According to generalized Snell’s law of reflection, when a plane wave is an incident vertically on an MS, the angle of reflection can be expressed as:(5)$$\theta_{r}=\sin^{-1}\frac{\lambda_{0}}{2\pi }\frac{\mathrm{d\varphi}}{\mathrm{dx}}=\sin^{-1}\left(\frac{\lambda_{0}}{L}\right)$$
where dφ/dx=2π/L, *L* represents the arrangement length of the unit corresponding to a reflection phase change period (360°). *L* = np, where ***p*** represents the lattice length of the element and n represents the number of elements corresponding to a period of reflection phase change.

Derived from Equation (5), the reflection phase gradient of MS elements is calculated as (reflection phase difference of adjacent elements):(6)$$△\varphi =\frac{2\pi p}{\lambda_{0}}\sin\theta_{r}$$

Based on the 1-bit coding MS, the design principle of the coded MS was analyzed, as shown in Figure 1. The MS is composed of *M × N* unit cells of the same size, and each unit can be “0” or “1” elements. Under normal incidence, the far-field scattering function *f*(*θ*,*φ*) can be expressed as:(7)$$f(\theta ,\varphi )=f_{c}(\theta ,\varphi )\sum_{m=1}^{N}\sum_{M=1}^{N}exp\left\{-i\left[\varphi \left(m,n\right)+kp\left(m-\frac{1}{2}\right)\sin\theta \cos\varphi +kp\left(n-\frac{1}{2}\right)\sin\theta \sin\varphi \right]\right\}$$
where $\varphi \left(m,n\right)$ are the reflection phase of the unit cell located at the position of [*m*, *n*], *θ* and ϕ are the elevation and azimuth angles of an arbitrary direction, respectively. When *θ* and φ satisfy the following conditions, the far field scattering amplitude reaches the first extreme value point:(8)$$\varphi =\pm \tan^{-1}\frac{{∆\varphi }_{y}}{{∆\varphi }_{x}}\frac{p_{y}}{p_{x}}\;\theta =\sin^{-1}\left(\frac{\lambda_{0}}{2\pi }\sqrt{{\left(\frac{{∆\varphi }_{x}}{p_{x}}\right)}^{2}+{\left(\frac{{∆\varphi }_{y}}{p_{y}}\right)}^{2}}\right)$$

Here, ${∆\varphi }_{x}$ and ${∆\varphi }_{y}$ are the phase differences of lattices along the x and y directions, respectively. In our 1-bit coded MS prototype, $p_{x}$ = $p_{y}$ = p, ${∆\varphi }_{x}$ = ${∆\varphi }_{y}$= 0/π. According to Equation (8), it can be known that when the units are all encoded as “0” or “1”, the reflected EM wave is a single beam opposite to the incident direction. The designed coding sequence “0101…/0101…” has dual-beam far-field scattering characteristics, and the coding sequence “0101…/1010…” has a quad-beam.

## 3. Results

As previously mentioned, applying various values of sheet resistances of graphene units at different locations across the MS array results in a reconfigurable coding layout. The different distribution of coding particles leads to diverse functionalities only achieved by one planar structure. In our work, three distinct graphene coding sequences are designed to achieve multiple-beam steering. Here, by changing the sheet resistances of graphene MS units to realize beam reconfiguration. As shown in Figure 3, the whole MS is occupied by 20 × 20 graphene unit cells of “0” and “1” units.

Figure 3a shows a coding sequence “00000000…/00000000…”, it has phase changes the ${∆\varphi }_{x}$ = 0 along the x-axis and ${∆\varphi }_{y}=$ 0 along the y-axis, respectively. Figure 3b shows a periodic coding sequence “01010101…/01010101…”; it has phase changes of ${∆\varphi }_{x}$ = ±π along the x-axis and ${∆\varphi }_{y}=$ 0 along the y-axis, respectively. Figure 3c shows a periodic coding sequence “01010101…/10101010…”, it has phase changes the ${∆\varphi }_{x}$ = ±π along the x-axis and ${∆\varphi }_{y}=$ ±π along the y-axis, respectively.

Figure 3d–f show the simulated far-field scattering patterns of the proposed graphene coding MSs by using the electromagnetic (EM) simulator CST Studio Suite, respectively. A normal x-polarized plane wave propagating along the z-direction illuminates all the MSs, and the corresponding far-field patterns at f = 55 GHz are depicted. The results show that the three different coding methods can generate single-beam, dual-beam, and quad-beam, respectively. The simulation results are consistent with the previous theoretical analysis.

In order to further verify the beam manipulation ability of the graphene-coded MS antenna, the precise phase distribution at a 30° reflection angle was calculated in MATLAB according to Equation (8), as shown in Figure 4a. The code is shown in Appendix A. Furthermore, radar cross-section reduction (RCS) can be reduced by forming random coding sequences. The random phase distribution Ψ to be integrated with the reflective MS is obtained by:Ψ = π/2 × randi([0, 1], 20, 20)(9)
where randi([0, 1], 20, 20) represents a 20 × 20 matrix with a value of pseudorandom distributed 0 or 1 for each cell, as shown in Figure 4b, the simulated far-field patterns at 55 GHz pointing at 30° are shown in Figure 4c,d. It can be seen that random MS sequences can scatter EM waves in more directions and realize the function of RCS.

Based on the proposed 1-bit unit cell and phase design, two 20 × 20 phased 1-bit GBCMs with beams pointing at 0° and 30° are designed with a center frequency of 55 GHz. Moreover, the simulated patterns in the xoz plane under normally incident waves at different frequencies are shown in Figure 5. The simulated beam directions of the 1-bit graphene MSs agree with the designed directions, and the measured SLLs in the xoz plane are suppressed below −10 dB.

## 4. Discussion

In this work, we propose a simple structured beam steering MMW antenna using novel GBCM, and its coding state can be manipulated by altering the sheet impedance of graphene instead of bias voltage. There have previously been reported many design schemes for GBCM antennas. Table 1 shows the differences and similarities between this work and previous studies.

Paper [20,25,26,27] proposed graphene MS structures to realize beam steering in THz waves by electrically reconfiguring the chemical potential of the graphene. Paper [28,29,30,31] achieved beam steering in the THz band by using different responses of graphene in different Fermi levels. In [32,33], the method of setting different bias voltages in the simulation is used to realize beam steering. However, all three methods essentially alter graphene’s EM properties by adding bias circuits, which can lead to complex structures that are difficult to manufacture with existing technology. Therefore, GBCM antennas are mostly in the simulation and numerical calculation stage. Fortunately, paper [34] fabricated and measured beam reconfiguration graphene antenna in microwaves by taking advantage of the tunable sheet impedances of graphene. However, it only implements simple beam generation and does not incorporate coding techniques to achieve more precise beam control. For the sake of simplicity in fabrication and improvement of beam manipulation, we designed GBCMs for the EM wave beam regulation and combined tunable sheet impedances with digital coding technology.

## 5. Conclusions

Here, for the first time, the broadband manipulation of MMW waves is acquired by introducing a multifunctional graphene-based coding metasurface (GBCM). The designed structure consists of subwavelength patterned graphene units whose operational statuses can be altered between two digital states of “0” and “1”. Its coding state can be manipulated by altering the sheet impedance of graphene instead of bias voltage. This results in a simplified design structure, providing possibilities for the subsequent fabrication of graphene MS antennas. In this work, the tunable behavior of graphene has led to various functionalities, among which we simulated five different coding sequences, implementing dual-, quad- and single-beam-generated beam deflection (30°), as well as a random coding sequence for radar cross-section (RCS) reduction.

## Figures and Tables

**Figure 1 micromachines-14-01018-f001:**
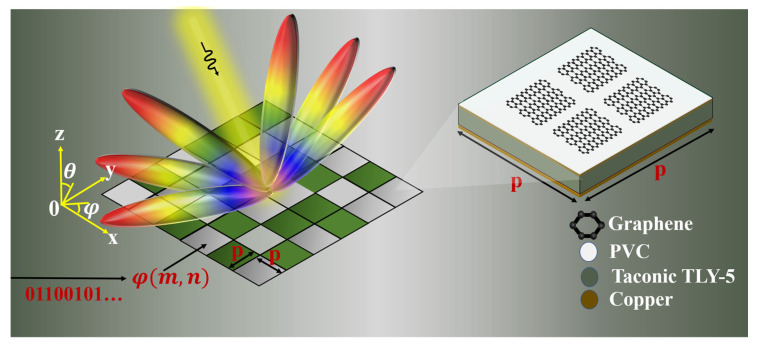
The geometry of coding MS and its constitutive graphene units.

**Figure 2 micromachines-14-01018-f002:**
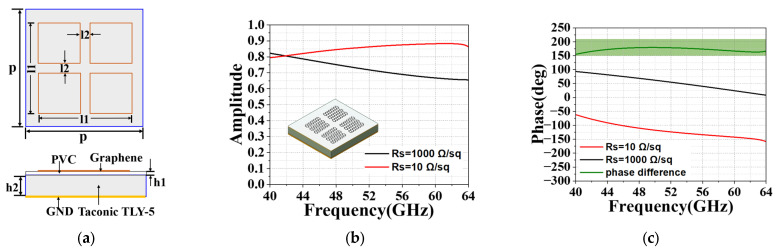
Schematic diagram of the proposed 1-bit graphene unit. (**a**) Top and side views, where *p* = 3.8 mm, *l*1 = 3.15 mm, *l*2 = 0.3 mm, *h*1 = 0.075 mm, and *h*2 = 0.787 mm. (**b**) Reflection amplitude. (**c**) Phases of the “0” and “1” elements with different sheet resistances.

**Figure 3 micromachines-14-01018-f003:**
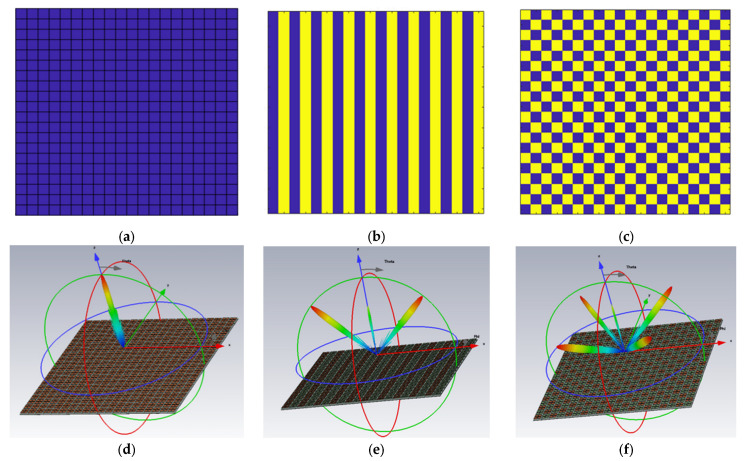
The simulation results of different (**a**–**c**) coding sequences in MATLAB and (**d**–**f**) far-field patterns of the proposed 1-bit graphene MSs at f = 55 GHz.

**Figure 4 micromachines-14-01018-f004:**
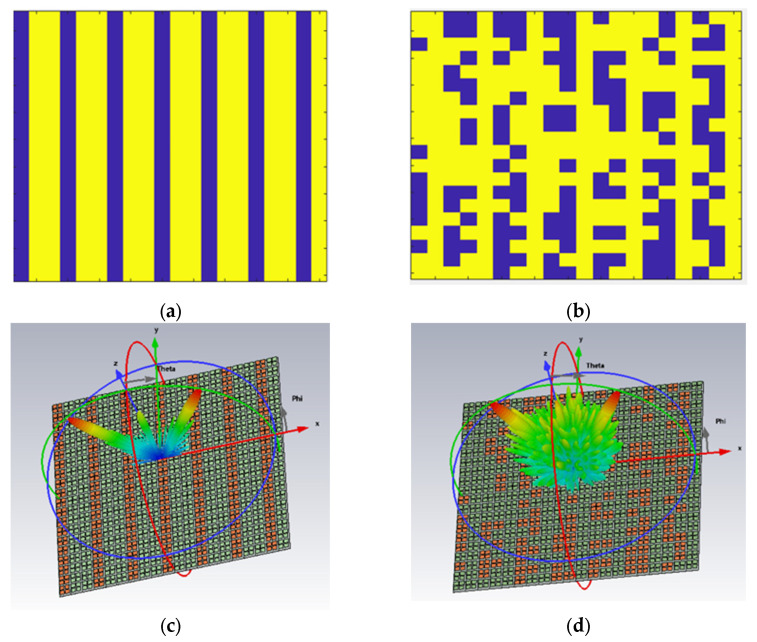
Calculated phase distributions and far-field scattering patterns for the 30° phase-compensated methods at f = 55 GHz under normally incident waves: (**a**) precise phase coding sequences; (**b**) random phase coding sequences; far-field patterns of the proposed 1-bit (**c**) precise; (**d**) random phase coding graphene MS.

**Figure 5 micromachines-14-01018-f005:**
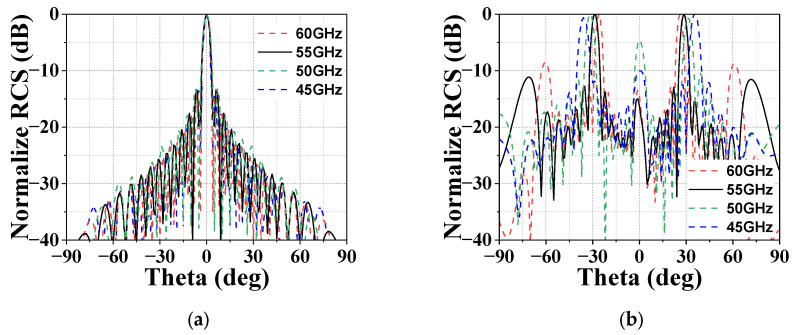
Simulated patterns of the random phase coding graphene MS under normally incident plane waves in the xoz plane: (**a**) 0° and (**b**) 30°.

**Table 1 micromachines-14-01018-t001:** Comparison of recently published GBCM antennas.

Ref.	Frequency	Unit Cell	Tunable Principle	Coding	Application
[20]	2 THz	Graphene patch	Chemical potential	2-bit	Beam steering
[25]	0.1 to 4 THz	Graphene patch	Chemical potential	2-bit	Absorption and reflection
[26]	1.5–1.9 THz	Graphene patch	Chemical potential	2-bit	Vortex beam generation
[27]	1–1.9 THz	Graphene patch	Chemical potential	1-bit	Scattering manipulation
[28]	1.0–1.4 THz	Graphene under meta-atom	Fermi level	8-bit	Multiple beam steering
[29]	1.15 THz	Graphene patch	Fermi level	≥2-bit	THz scattering
[30]	0.8–1.6 THz	Graphene patch	Fermi level	1-bit	Dynamic beam steering
[31]	1.5 THz	Graphene microribbons	Fermi level	1-bit	Secure communication
[32]	12–16 GHz	Graphene on metallic resonators	Bias voltage	1-bit	Scattering steering
[33]	-	Graphene patch	Bias voltage	No	Beam scanning
[34]	13 GHz	Graphene ribbon	Sheet resistance	No	Beam reconfiguration
This work	55 GHz	Graphene patch	Sheet resistance	1-bit	Beam steering

## Data Availability

Not applicable.

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
