# Peer review of "Multiple-Beam Steering Using Graphene-Based Coding Metasurfaces"

_micromachines, 2023, doi:10.3390/mi14051018_

Round 1

Reviewer 1 Report

The authors have demonstrated their success in modeling and designing a reconfigurable millimeter-wave (MMW) antenna using a graphene-based coding metasurface (GBCM). In their study, they presented a 1-bit coding method for beam steering based on the sheet resistance principle. To enhance the manuscript further, I recommend the authors to address the following questions/suggestions:

  1. The manuscript would benefit from the inclusion of equivalent circuit models to illustrate the proposed device.
  2. The algorithm used for coding should be listed in the appendix of the manuscript.

With these minor revisions, I recommend accepting the manuscript.

Reviewer 2 Report

In this article, the authors propose a simple structured beam reconfigurable millimeter wave (MMW) antenna using a novel graphene encoded metasurface (GBCM). Different from the previous method, its coding state can be manipulated by altering the sheet impedance of graphene instead of bias voltage. The theoretical and simulation results show that graphene has great potential for MMW manipulation applications which lay a foundation for the subsequent development and fabrication of GBCM. I believe that publication of the manuscript may be considered only after the following issues have been resolved.

1. The article mentions that the encoding state can be achieved by changing the thin layer impedance of graphene. How did the authors achieve this?

2. What method or software was used to simulate this work? And, what is the thickness of graphene in this work?

3. In the model in this work, the author mentioned some specific parameters. May I ask if this structural parameter has any impact on the final performance? The authors need to provide some criteria.

4. The introduction can be improved. The articles related to some applications of graphene materials should be added such as Results in Physics 48, 2023, 106420; Diamond & Related Materials 128 (2022) 109273; Optics Express, 30(20), 35554-35566, 2022; Talanta 2015, 134, 435–442.

5. Please check the grammar and spelling mistakes of the whole manuscript.

Minor editing of English language required

Round 2

Reviewer 2 Report

Accept in present form.